# Coronary Plaque Regression and Fractional Flow Reserve Improvement in a Chronic Coronary Syndrome Case: Early Optimal Medical Therapy and Fractional Flow Reserve-Computed Tomography Follow-Up Strategy

**DOI:** 10.3390/diseases12110297

**Published:** 2024-11-20

**Authors:** Yuki Yoshimitsu, Toru Awaya, Naoyuki Kawagoe, Taeko Kunimasa, Raisuke Iijima, Hidehiko Hara

**Affiliations:** 1Department of Cardiovascular Medicine, Toho University Ohashi Medical Center, 2-22-36, Ohashi Meguro-ku, Tokyo 153-8515, Japan; 2Division of Diabetes, Metabolism and Endocrinology, Department of Internal Medicine, Toho University Ohashi Medical Center, Tokyo 153-8515, Japan

**Keywords:** optimal medical therapy, fractional flow reserve-computed tomography, chronic coronary syndrome, legacy effect, percutaneous coronary intervention

## Abstract

**Background:** Optimal medical therapy (OMT) is increasingly recognized as a cornerstone in managing chronic coronary syndrome (CCS), offering a non-invasive alternative to percutaneous coronary intervention (PCI). **Case Presentation:** A 38-year-old male with diabetes, dyslipidemia, and hypertension was treated with early and comprehensive OMT, including statins, ezetimibe, sodium-glucose cotransporter 2 inhibitors (SGLT2i), pioglitazone, and renin-angiotensin system inhibitors. Insulin was introduced during the acute phase to stabilize glycemic control. His HbA1c decreased to 6.3% within 4 months. **Results:** Over 8 months, the patient experienced a reduction in coronary plaque burden and an improvement in fractional flow reserve (FFR) from 0.75 to 0.90, indicating enhanced coronary blood flow. Plaque volume burden decreased from 85% to 52% in key coronary segments. **Conclusions:** This case highlights the effectiveness of OMT, including statins, ezetimibe, SGLT2i, and pioglitazone, in achieving outcomes comparable to PCI. FFR-computed tomography follow-up is critical in guiding treatment decisions. Continued OMT is recommended if plaque stabilization is observed. If no improvement is observed, OMT should be intensified, and PCI considered as appropriate.

## 1. Introduction

Chronic coronary syndrome (CCS) remains a major global health concern, contributing significantly to cardiovascular morbidity and mortality [1]. While percutaneous coronary intervention (PCI) has traditionally been a primary treatment approach for patients with coronary artery disease (CAD), recent evidence highlights the role of optimal medical therapy (OMT) in achieving outcomes comparable to PCI, particularly in CCS patients, as shown in the ISCHEMIA trial (Initial Invasive or Conservative Strategy for Stable Coronary Disease) [2]. In the SCOT-HEART (Scottish Computed Tomography of the Heart) trial, the addition of coronary computed tomography (CT) imaging to standard care reduced mortality and increased the initiation of preventive therapies, including statins [3].

Coronary CT imaging is now recommended as a first-line non-invasive diagnostic modality for suspected CCS, according to the current standards of the 2024 European Society of Cardiology (ESC) guidelines [1]. Coronary CT imaging is particularly indicated for patients with a low-to-moderate pre-test likelihood (5–50%) of obstructive CAD. This modality provides accurate anatomical assessment of coronary stenosis, with high sensitivity and specificity for excluding CAD compared to invasive coronary angiography (ICA). This allows clinicians, especially in cases of atypical angina, to avoid the risks associated with invasive diagnostics and prioritize ICA for patients with a higher likelihood of needing intervention.

Advanced imaging technologies, such as fractional flow reserve-CT (FFR-CT), are essential for guiding treatment decisions, including the need for PCI [4]. FFR-CT combines routine CT data with computational fluid dynamics to assess the hemodynamic impact of coronary stenosis by estimating coronary blood flow and calculating FFR values, enabling a non-invasive evaluation [5].

Recent advancements in pharmacological therapies, as outlined in the 2024 ESC guidelines, have significantly broadened the options for managing dyslipidemia, diabetes, and other cardiovascular risk factors in patients with CCS. In particular, lipid-lowering therapies, including statins, ezetimibe, proprotein convertase subtilisin/kexin type 9 inhibitors (PCSK9i), and bempedoic acid, alongside emerging agents such as sodium–glucose cotransporter 2 inhibitors (SGLT2i) and glucagon-like peptide-1 receptor agonists (GLP-1 RA), have shown significant benefits in reducing the incidence of cardiovascular events [1].

Incorporating these diagnostic and therapeutic advancements, the management of CCS now emphasizes a dynamic approach that prioritizes long-term cardiovascular risk reduction, often referred to as the “legacy effect”. The legacy effect refers to the long-term cardiovascular benefits achieved through early and sustained intervention, particularly in managing glycemic levels and aggressively lowering low-density lipoprotein cholesterol (LDL-C) levels. This early intervention has been shown to reduce the incidence of future cardiovascular events even after treatment goals have been met [6,7].

## 2. Case Report

A 38-year-old male (height: 166 cm; weight: 82 kg; body mass index (BMI): 30 kg/m^2^) was admitted for insulin initiation and lifestyle changes because of worsening diabetes mellitus (HbA1c = 12.8%). He also had dyslipidemia and hypertension, with an LDL-C level of 154 mg/dL, a high-density lipoprotein cholesterol (HDL-C) level of 49 mg/dL, triglyceride (TG) level of 416 mg/dL, and systolic blood pressure of 180 mmHg. He experienced chest discomfort, and his electrocardiogram revealed ST segment depression in leads I, II, aVL, V5, and V6. Coronary computed tomography (CT) (Aquilion ONE, Canon Medical Systems, Otawara, Japan) revealed moderate stenosis in the mid-left anterior descending artery (LAD) with an FFR of 0.75 (Figure 1A).

Since the FFR was below 0.8, both PCI and OMT were viable options for CCS. The patient chose OMT after discussing the potential benefits and risks of each approach. Initiation of insulin and oral medications (dapagliflozin 10 mg, pioglitazone 30 mg, and metformin 500 mg) improved his HbA1c to approximately 8% within 2 months. After switching to oral medications, the patient’s HbA1c decreased to 6.3% within 4 months and remained stable between 6.2% and 7.4% thereafter. For dyslipidemia, pitavastatin 4 mg, ezetimibe 10 mg, and omega-3 fatty acids 2 g were introduced. For hypertension, 40 mg of olmesartan and 20 mg of nifedipine were administered. Aspirin was also introduced for CCS. The LDL-C level of the patient was 30–66 mg/dL. A reduction in LDL-C from baseline, from 57% to 81%, was observed. The patient’s blood pressure also stabilized around 140 mm Hg. Additionally, his total lipoprotein(a) (Lp(a)) decreased from 37.2 to 11.8 mg/dL, and his CRP level decreased from 0.08 to 0.03 mg/dL.

A follow-up CT performed after 8 months revealed an improvement in stenosis and an increase in the FFR from 0.75 to 0.90, indicating a significant enhancement in coronary blood flow (Figure 1A,B). After treatment, the plaque volume burden was reduced from 78% to 59% on segment 2 and from 85% to 52% on segment 3. The minimum lumen area on segment 3 improved from 2.91 to 5.37 mm^2^. Additionally, there was an improvement in CT values, indicating plaque stabilization.

(A-1, 2, 3) The mid-LAD artery plaque volume burden was 85%, 78%, and 85%, respectively, before treatment. (B-1, 2, 3) After treatment, the plaque volume burden improved to 81%, 59%, and 52%, respectively. The minimum lumen area also improved from 3.08 to 3.76, 4.41 to 5.44, and 2.91 to 5.37 mm^2^, respectively. (A and B-1, 2, 3) The mean plaque CT value improved from 25 to 34, 50 to 64, and 35 to 52 HU, respectively, suggesting plaque stabilization. Plaques with a CT value below 30 HU are classified as low-attenuation plaques, which are associated with higher risk, and the observed improvements indicate a shift away from this more vulnerable plaque type.

FFR, fractional flow reserve; CTA, computed tomography angiography; LAD, left anterior descending artery; CT, computed tomography; HU, Hounsfield units.

## 3. Discussion

In this case, the patient’s HbA1c decreased to 6.3% within four months, and intensive LDL-C management was also implemented early on. Both interventions are likely to contribute to the “legacy effect”, which refers to long-term cardiovascular risk protection [6,7]. This case illustrates that early, aggressive LDL-C reduction not only leads to notable plaque regression but also contributes to the stabilization of atherosclerotic plaques, a key factor in reducing the incidence of future cardiovascular events [8,9,10]. This highlights the potential of OMT to achieve outcomes that are comparable to PCI.

FFR-CT has emerged as a non-invasive tool, offering both anatomical and functional insights. The PLATFORM (Prospective Longitudinal Trial of FFR_CT_: Outcome and Resource Impacts) trial showed that FFR-CT reduces unnecessary invasive coronary angiography [4]. In this case, early OMT combined with FFR-CT follow-up confirmed both plaque regression and improved coronary flow. In recent years, fluid dynamics analysis using FFR-CT has enabled the measurement of wall shear stress, predicting the risk of progression from CCS to acute coronary syndrome (ACS). This approach allows for more precise risk assessment by incorporating not only anatomical evaluations but also hemodynamic risk factors. Furthermore, it is important to investigate whether OMT can improve not only plaque stabilization but also shear stress, which could play a critical role in future therapeutic strategies [11,12].

There has been a recent rise in the number of medications that reduce the incidence of cardiovascular events in patients with dyslipidemia and diabetes. Examples include PCSK9i, SGLT2i, and GLP-1 RA [13,14]. These medications have allowed for the maintenance of outcomes equivalent to PCI for CCS patients with moderate or severe ischemia. The ISCHEMIA trial, in which 40% of the study population had diabetes, demonstrated the equivalent efficacy of OMT and PCI [2]. The present case showed that OMT using statins, ezetimibe, SGLT2i, thiazolidinedione (TZD), and eicosapentaenoic acid (EPA) leads to plaque regression and achieves effects equivalent to those of PCI [2,13].

The legacy effect suggests that early interventions help to achieve glycemic control besides lowering LDL-C, contributing to a long-term reduction in cardiovascular event risk. In the comparison of HbA1c levels during the initial 0-to-1-year period, those who achieve HbA1c < 6.5% have significantly lower rates of microvascular and macrovascular events than those with HbA1c ≥ 6.5%. In this case, HbA1c levels dropped to 6.3% within just four months, underscoring the effectiveness of early intervention [6]. Early initiation of PCSK9i, such as evolocumab, significantly reduces the incidence of long-term cardiovascular events [7]. This effect is attributed to the aggressive lowering of LDL-C, leading to plaque regression and increased fibrous-cap thickness, which plays a crucial role in stabilizing atherosclerotic plaques [9,10]. Therefore, the legacy effect may be enhanced by early management of both glycemic levels and LDL-C, providing sustained cardiovascular protection.

SGLT2i, along with GLP-1 RA, are classified as Class 1 therapies for patients with type 2 diabetes mellitus and atherosclerotic cardiovascular disease. The effect of these therapies extends beyond glycemic control, encompassing the reduction in blood pressure, body weight, and TG levels [13]. Specifically, SGLT2i have been shown to suppress the activity of the NLRP3 inflammasome, a critical component for the production of pro-inflammatory cytokines such as interleukin (IL)-1β and IL-18 within macrophages [15]. These cytokines play a significant role in the progression and potential rupture of atherosclerotic plaques, which are major contributors to coronary artery disease and related cardiovascular events. Furthermore, the increase in urinary angiotensin-converting enzyme 2 (ACE2) observed with SGLT2i may also contribute to its anti-inflammatory effects [16], as ACE2 facilitates the conversion of angiotensin II to angiotensin (1–7), promoting vasodilation and reducing inflammation within the kidneys. This mechanism suggests an additional pathway by which SGLT2i could mitigate renal inflammation and protect renal function [17]. The GLP-1 RA should be considered for patients with CCS who do not have diabetes but are overweight or obese (BMI > 27 kg/m^2^), as it has been shown in the SELECT (Semaglutide Effects on Cardiovascular Outcomes in People With Overweight or Obesity) trial to reduce the risk of cardiovascular mortality or stroke [1,18].

TZDs, particularly pioglitazone, are also known for reducing the incidence of cardiovascular events. The impact of pioglitazone on vascular endothelial function is reported to be comparable to that achieved with SGLT2i and GLP-1 RA [14]. Pioglitazone improves vascular endothelial function by reducing oxidative stress and improving endothelial nitric oxide availability, contributing to better vascular reactivity. The PERISCOPE study (Pioglitazone Effect on Regression of Intravascular Sonographic Coronary Obstruction Prospective Evaluation) showed that pioglitazone was associated with coronary plaque regression [19], while the PROactive study (PROspective pioglitAzone Clinical Trial In macroVascular Events) demonstrated a reduction in the incidence of cardiovascular events over a 10-year follow-up period [20]. However, TZDs may also exacerbate the risk of heart failure. This adverse effect is primarily because of fluid retention, associated with the increased mRNA expression of aquaporin (AQP)2 and AQP3 water channels in the renal medulla [21]. However, when combined with SGLT2i, the risk of fluid retention and the associated increase in myocardial weight is mitigated [21]. Therefore, a combination of SGLT2i and TZD has been reported to effectively reduce the risk of cardiovascular events and heart failure [22]. This dual risk reduction is hypothesized to result from SGLT2i’s ability to attenuate acute heart failure episodes in the short term, whereas TZD reduces the risk of coronary artery disease, thereby decreasing the incidence of ischemic heart failure in the long term.

Statin therapy allows patients to achieve low LDL-C levels (<55–70 mg/dL), and a reduction in LDL-C by ≥50% from baseline is associated with both plaque regression and a reduction in the incidence of cardiovascular events [23]. Furthermore, it is important not only to lower LDL-C levels but also to start statin therapy early. The ESCORT (Effect of Early PitavaStatin Therapy on Coronary Fibrous-cap Thickness assessed by Fourier-Domain Optical CoheRence Tomography) study reported that early therapy with pitavastatin for patients with ACS increased fibrous-cap thickness, as assessed via optical coherence tomography, compared to the group in which therapy was started after 3 weeks [8]. Similarly, statin therapy has been reported to result in plaque regression, as assessed using coronary CT [24]. The IMPROVE-IT (IMProved Reduction of Outcomes: Vytorin Efficacy International Trial) showed that ezetimibe combined with a statin is more effective in lowering LDL-C levels below 55 mg/dL than statin therapy alone and achieving a significant reduction in the incidence of cardiovascular events, particularly in diabetic patients, similar to the present case [13].

Bempedoic acid is a Class 1 recommendation in recent guidelines for patients who are statin intolerant and do not achieve their target LDL-C levels with ezetimibe [1]. It provides a safe and effective therapeutic option for lowering LDL-C levels while avoiding the adverse effects associated with statins, particularly muscle pain. Additionally, bempedoic acid exhibits anti-inflammatory effects, and the CLEAR (Cholesterol Lowering via Bempedoic Acid, an ATP Citrate Lyase (ACL)–Inhibiting Regimen) Outcomes trial has confirmed its efficacy in reducing the incidence of cardiovascular events [25].

EPA, a major component of omega-3 fatty acids, was shown in the REDUCE-IT (Reduction of Cardiovascular Events with Icosapent Ethyl–Intervention) trial to significantly reduce the risk of cardiovascular events in statin-treated patients with high cardiovascular risk and elevated TG levels. The trial highlighted that EPA reduced these risks in patients with and without diabetes, regardless of Lp(a) levels [26]. In this case, however, an omega-3 fatty acid containing both EPA and docosahexaenoic acid (DHA) was used. Although DHA has been shown to have additional anti-atherosclerotic effects when combined with EPA [27], a recent study evaluating coronary plaque using cardiac magnetic resonance found no significant difference, meaning that the effects of EPA/DHA remain controversial [28]. Interestingly, EPA/DHA have also been reported to activate peroxisome proliferator-activated receptor (PPAR)α, promoting lipid metabolism and anti-inflammatory effects. When combined with TZDs, which are agonists of PPARγ, EPA/DHA may further enhance insulin sensitivity and promote adipocyte differentiation. These potential synergistic effects warrant further investigation to better understand the combined therapeutic benefits of EPA/DHA and TZDs [29].

Lp(a) decreased from 37.2 mg/dL to 11.8 mg/dL in this case, which is considered to be an effect of both pioglitazone and EPA. In a comparative study between pioglitazone and rosiglitazone, the Lp(a) reduction was −19.7% in the pioglitazone group versus 0.5% in the rosiglitazone group [30]. This reduction is thought to be due to pioglitazone’s influence on not only PPARγ but also PPARα [31]. Additionally, there is evidence indicating a 5% reduction in Lp(a) levels with EPA [32]. PPARα activity may contribute to the reduction in Lp(a) levels through enhanced fatty acid oxidation and TG reduction [33]. Statins generally have no clinically significant effect on Lp(a) levels. However, moderate-intensity pitavastatin has been observed to have a slight, though statistically insignificant, effect in reducing Lp(a) levels [34]. In this case, the use of pitavastatin may have contributed to the reduction in Lp(a) levels. Antisense oligonucleotides and small interfering RNA therapies significantly reduce Lp(a) levels by inhibiting the formation of apo(a), a key component of Lp(a). These therapies are currently under investigation in clinical trials [35].

Lifestyle adherence is essential for the full realization of OMT’s benefits in managing CCS. The ESC guidelines highlight that without sustained lifestyle modifications, the impact of OMT may be limited. To enhance adherence, simplifying medication regimens through fixed-dose combinations, and involving multiprofessional teams, family members, and patients themselves in the treatment plan, is strongly recommended [1].

## 4. Conclusions

Early and comprehensive initiation of OMT can reduce coronary plaque burden and improve FFR, potentially decreasing the need for PCI. This case highlights the importance of FFR-CT follow-up for treatment decisions. When FFR-CT follow-up shows improvement, as seen in this case, it is advisable to continue OMT. However, if no improvement is observed or if progression is detected, it may be necessary to supplement OMT with PCI. This approach offers a strategy for managing CCS, emphasizing the critical role of timely and tailored therapy.

## Figures and Tables

**Figure 1 diseases-12-00297-f001:**
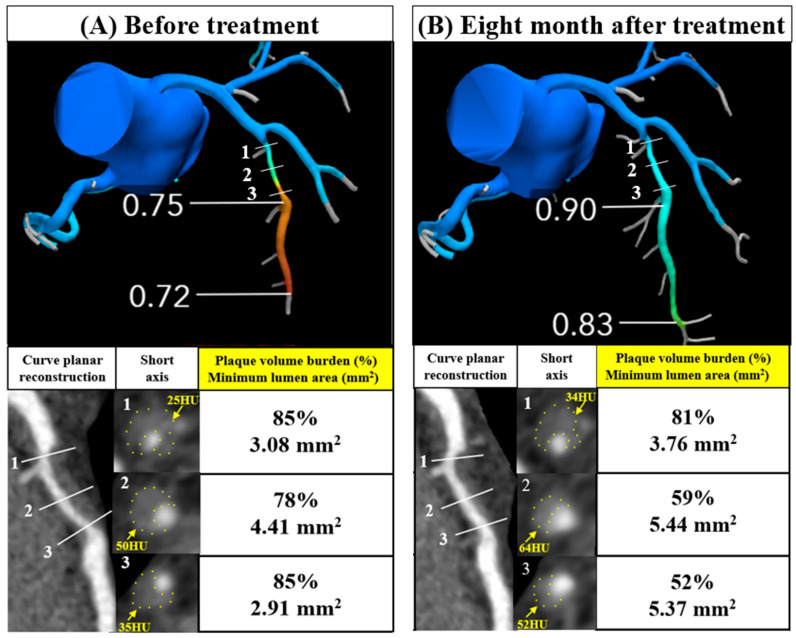
Changes in FFR and plaque volume evaluations in coronary CTA before treatment and at the 8-month follow-up visit (**A**) The mid-LAD artery showed moderate stenosis with an FFR of 0.75. (**B**) The degree of stenosis and FFR improved to 0.90. The improvement in FFR from 0.75 to 0.90 indicates a significant enhancement in coronary blood flow. The heart rates during each CT scan were 59 and 61 beats per minute, respectively.

## Data Availability

No new data were created or analyzed in this study. Data sharing is not applicable to this article.

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
