# Peer review of "Coronary Plaque Regression and Fractional Flow Reserve Improvement in a Chronic Coronary Syndrome Case: Early Optimal Medical Therapy and Fractional Flow Reserve-Computed Tomography Follow-Up Strategy"

_diseases, 2024, doi:10.3390/diseases12110297_

Round 1

Reviewer 1 Report

Comments and Suggestions for Authors

Authors presented an interesting case of young man with CHD that demonstrated possibility of the best and adequate medical therapy to regress coronary plaque volume. The paper is well written and illustrated. The methods of evaluation are adequate and supported the study results. The significant reduction of LDL-C is anticipated on therapy with pitavastatin and ezetimibe as well as TGs reduction on PUFA. 

I have no major concerns.

Minor:

1. What type of PUFA was administered?

2. Lp(a) level was reduced by more than 50%. Authors should explain by what mechanisms? Also they mentioned Lp(a)-cholesterol but not total Lp(a) level. They should clarify the method of measurement.

Reviewer 2 Report

Comments and Suggestions for Authors

A diagnostic method that is gradually gaining importance in the diagnosis of ischemic heart disease (IHD) is computed tomography of coronary arteries (CCTA). According to the standards of the American Heart Association (AHA) and the European Society of Cardiology (ESC), CCTA is currently recommended as one of the basic diagnostic methods in patients with suspected CCS, as an alternative method to invasive coronary angiography (ICA). This case reprot very meticulously and in detail describes the effectiveness of OMT, including statins, ezetimibe, SGLT2i, and pioglitazone, in achieving outcomes comparable to PCI and underscores the importance of FFR-CCTA follow-up in guiding treatment decisions.

Manuscript is well written. Nevertheless, to become suitable for publication, the Authors should raise the manuscript profile adding some relevant additional details and comments.

1. In the introduction, it is worth mentioning and citing relevant documents that, according to the standards of the American Heart Association (AHA) and the European Society of Cardiology (ESC), CCTA is currently recommended as one of the basic diagnostic methods in patients with suspected CCS, as an alternative to invasive coronary angiography (ICA). CCTA has proven diagnostic accuracy in the anatomical assessment of stenoses in coronary arteries, and particularly high sensitivity and specificity in excluding the presence of stenoses compared to ICA, which allows, especially in the group of patients with atypical angina, to abandon the still risky invasive diagnostics and free up space for patients who really require such diagnostics.

2. The FFR-CCTA methodology should be described to familiarize the reader.

3. It is also advisable, in the case description, to consider the parameters of inflammation, especially hsCRP, because atherosclerosis is known to have an inflammatory background, and destabilization of the atherosclerotic plaque is accompanied by increased inflammation. Conversely, a reduction in the volume of atherosclerotic plaque and its stabilization, as described in the presented case, is usually associated with a reduction in the pro-inflammatory state.

Notwithstanding the foregoing, the work is valuable and very innovative. I my opinion this paper deserves to be published in the pages of the Diseases. Only minor changes are necessary.

Reviewer 3 Report

Comments and Suggestions for Authors

Congratulations on the work done and on the manuscript.

I suggest that the introduction frame the importance of a healthy lifestyle for a more sustained reduction in cardiovascular risk and for an even lower probability of an acute cardiovascular event.

Although OMT is an excellent option that can can help avoid coronary angiography and subsequent PCI, if the patient does not effectively change their behaviors in the long term and adopt a healthy lifestyle, the beneficial effects of OMT will be very limited. And as such, I believe that this problematic should be addressed in the introduction and in the conclusion / implications for future practice or research
